# No Detectable Effects of Chronic Stress on Moral Decision-Making Are Found: A Data Reanalysis and a Pre-Registered Replication and Extension

**DOI:** 10.3390/bs15081068

**Published:** 2025-08-06

**Authors:** Lemei Zou, Junhong Wu, Chuanjun Liu

**Affiliations:** 1Department of Sociology and Psychology, School of Public Administration, Sichuan University, Chengdu 610065, China; 2023225015043@stu.scu.edu.cn (L.Z.);; 2Institute of Psychology, Sichuan University, Chengdu 610065, China

**Keywords:** chronic stress, moral dilemma, CAN algorithm, CNI model

## Abstract

According to the dual-process model of moral cognition, individuals tend to refuse the proposal of killing some to save more innocents under stressful conditions compared to non-stressful states, which has been demonstrated in previous studies. However, this effect might be unreliable according to the General Adaptation Syndrome theory and the Stress Process Model. To test this speculation, we reanalyzed the raw data on the effect of chronic stress on moral choice from a previous study (Study 1) and conducted a pre-registered replication and extension study (Study 2). Both results demonstrated no detectable effect of chronic stress on moral decisions, which is inconsistent with the original results. This study calls for caution regarding this effect and has important theoretical and practical implications.

## 1. Introduction

In a fast-paced society, stress pervades daily life. People constantly endure pressure from various sources, such as interpersonal relationships and economic burdens ([18]). Chronic stress refers to the sustained psychological and physiological responses elicited by prolonged or repeated exposure to unpredictable or uncontrollable stressors ([7]; [27]). In contrast to acute stress, which is episodic and stems from specific stimuli, chronic stress, due to its recurring and persistent nature ([6]), exerts more extensive effects on individuals’ psychological and behavioral patterns. A phenomenon among these effects worthy of attention is the influence of chronic stress on decision-making behavior and the quality of decisions made ([43]).

Moral psychologists have increasingly focused on the influence of stress on moral decision-making. Researchers have discovered that inducing stress in individuals tends to make them reject proposals that involve harming one innocent person to save five others ([43]; [45]). This decision-making scenario originates from a classic paradigm in moral dilemmas known as the trolley problem, originally proposed by [15] ([15]).

An oncoming train is about to kill five railway workers. You have the option to switch the track, but by doing so, the train will kill one worker on the other track. Would you switch the track in such a situation?

In such moral dilemmas, researchers have categorized moral judgments into rule-based (deontology) moral judgments and outcome-based (utilitarianism) moral judgments ([46]). Rule-based moral judgments emphasize norm adherence and regard actions unacceptable if they violate norms, irrespective of potential benefits. For instance, not switching the track in the trolley problem aligns with deontology. Outcome-based judgments prioritize maximizing benefits, considering harmful acts morally acceptable if they bring greater well-being. Switching the track in the trolley problem exemplifies utilitarianism.

[45] ([45]) demonstrated that inducing stress in individuals leads to fewer utilitarian judgments. Later, scholars used the multinomial processing tree approach to separate consequence-driven, norm-driven and generalized inaction/action-driven processes in moral dilemma decisions (short as the CNI model). It was revealed that chronic stress heightens the tendency for inaction, indicating that chronic stress leads to an increased preference for deontological decision-making ([47]). However, since chronic stress develops gradually and its stressors remain concealed, its influence on moral decision-making remains uncertain. To address this question, the researchers reviewed prior studies on social stressors.

### The Influence of Stress on Moral Decision-Making

Research on social stressors began in mid-1950s ([41]) and expanded thereafter, with the three most common sources being life events, chronic stress, and daily hassles. Previous studies have found that although social stressors have negative impacts on health ([3]; [9]; [23]), their effects can be mitigated by other factors. [8] ([8]) discovered that social support, as a psychological resource, plays a role in alleviating stress, thus weakening the negative impact of stress. [14] ([14]) further emphasized that the stress process constitutes a dynamic system in which responses to stressors are moderated by situational difficulty, personal resources, and personality traits. Building on extant theory, [18] ([18]) demonstrated empirically that stress responses are jointly determined by external demands and personal resources. External demands denote challenges and tasks that function as potential stressors, whereas resources encompass internal or external supports, such as emotion regulation capacity, social support, and financial assets, which are available for coping with those demands. A higher demand-to-resource ratio (i.e., resource insufficiency) significantly amplifies perceived stress, as indexed by elevated blood pressure ([18]). In other words, as early as the end of the 20th century, scholars had discovered that the physiological and psychological impacts of stressors (especially chronic stress and life events) on individuals are relatively weak because mitigating factors enable the organism to have more adequate resources to cope with such stress. Therefore, it is plausible to speculate that chronic stress may exert limited influence on decision-making, particularly in moral decision-making.

Both the General Adaptation Syndrome (GAS) theory ([40]) and the stress process framework ([34]) support the view that chronic stress has limited effects on individuals. GAS posits three stages of stress response: alarm, resistance, and exhaustion. In the alarm stage, individuals closely monitor changes in the environment and activate their defense systems. During the resistance stage, individuals counterbalance external stress through internal defensive forces, restoring a balance between their mind and body. In the exhaustion stage, individuals’ resources are depleted, leading to psychological disorders and a cascade of stress-related adverse outcomes. [36] ([36]) also pointed out that the response to chronic stress is actually a defense mechanism, with adaptation at its core. Although the stressor remains, individuals develop adaptive capacity, thus weakening its impact on their physical and mental health. Similarly to GAS, the stress process framework argues that the stress process consists of three main components: stressors, resources, and outcomes. It highlights that stress outcomes depend on the interplay between stressors, resources, and outcomes, implying that individuals can utilize resources to mitigate the negative effects of stress ([34]). Both of the aforementioned theoretical models emphasize the organism’s adaptability to stress. When exposed to stress, individuals can regulate and mitigate its negative effects through psychological and social resources such as social support and coping strategies. It is only when an individual lacks sufficient supportive resources or physiological defenses to regulate the stressor that they will be further negatively impacted by stress.

Given the stable and long-term presence of chronic stress, individuals are more likely to develop corresponding coping mechanisms to alleviate the impact of these stressors on their physical and mental health. Although some studies suggest that chronic stress affects individual behavioral decision-making, [35] ([35]) emphasized that the impact of stress on cognitive processes is reversible, and individuals experiencing chronic stress still have the opportunity to recover from its negative effects. This view is also supported by research conducted by [42] ([42]). [42] ([42]) found that individuals under chronic stress exhibit behavioral tendencies toward automated reactions when performing decision-making tasks, because people become less sensitive to the value of the outcome under chronic stress. Furthermore, fMRI findings revealed that brain function and structural changes induced by chronic stress are reversible; after 6–7 weeks of stress alleviation, decision-making behavior and related brain networks can return to normal ([42]). Therefore, it is reasonable to speculate that the negative impact of chronic stress on individuals is not as strong as previously assumed. It is necessary for researchers to conduct further research on the relationship between chronic stress and moral decision-making.

Not only from a theoretical perspective but also from a methodological one, the conclusion that chronic stress affects moral decision-making warrants cautious re-examination. [47] ([47]) reported that chronic stress increases deontological decision-making tendencies, a finding potentially shaped by their methodological approach, which employed the CNI model to quantify moral decision-making tendencies. However, the CNI model has been criticized for its numerous methodological limitations ([4], [5]; [13]; [29], [30]), which may increase the risk of false positives in research conclusions ([12]; [31]). These limitations have been addressed in the subsequent CAN algorithm ([29]). The CAN algorithm utilizes algebraic computation strategy and introduces more parameters to measure the various possibilities in people’s moral decision-making processes ([29]). Therefore, based on the methodological limitations of the CNI model and the relevant theoretical perspectives of stress, it is necessary to employ the relatively refined CAN algorithm to test how chronic stress affects moral decision-making, thereby clarifying the effects of chronic stress on moral decision-making.

The current research study aims to reassess the impact of chronic stress on moral decision-making. In Study 1, the researchers will utilize the CAN algorithm to reanalyze the original data from [47] ([47]). To further test the robustness of the research results, Study 2 replicates the study and further incorporates relevant control variables to evaluate the existence of this effect.

## 2. Study 1: Data Reanalysis of Moral Decision-Making Under Chronic Stress

### 2.1. Methods and Materials

By contacting the original authors, we obtained the raw data from [47] ([47]). In the original data, [47] ([47]) recruited 197 undergraduate students from psychology courses (65.5% female, 34.4% male; age range 18–22 years, M = 19.49, SD = 0.83).

For this study, the measurement of chronic stress adopted the 10-item Perceived Stress Scale (PSS-10) translated by [44] ([44]). The scale uses a five-point Likert scale ranging from 0 (never) to 4 (very often). The total score ranges from 0 to 40, with a higher score indicating greater perceived chronic stress over the past month (Cronbach’s α = 0.80). The moral decision-making materials were selected from the 24 moral scenario stories compiled by [17] ([17]). Participants rated their acceptance of behaviors on a 1-6 scale (1 = completely disagree, 6 = completely agree). In the CNI model analysis, participants were divided into a high-stress group (M = 31.14, SD = 3.28, N = 92) and a low-stress group (M = 22.45, SD = 2.91, N = 105) based on their PSS-10 scores, using the median-split method.

In the first part, following the analysis strategy of the original authors, the researchers recoded the responses of moral dilemmas, with “4–6” indicating approval and “1–3” indicating disapproval. As previous studies demonstrated that moral dilemma decisions had gender differences ([16]), the researchers controlled for gender as a covariate. Subsequently, the CAN algorithm was employed to calculate three core decision-making parameters: consequence sensitivity (C parameter), norm sensitivity (N parameter), and overall action/inaction bias (A parameter) ([29]). Furthermore, considering that the parameters in the CAN algorithm can be linearly correlated, the researchers calculated the correlations between chronic stress levels and the parameters in moral decision-making, rather than comparing the differences between the artificially divided high-stress and low-stress groups.

In the second part, the researchers recalculated moral decision parameters using original continuous data to avoid measurement errors from categorization ([1]; [2]). The researchers analyzed correlations between perceived stress levels and moral decision parameters using original PSS scores.

### 2.2. Results

As shown in Table 1, the reanalysis of dichotomized moral decision data indicates that, after controlling for gender, chronic stress is negatively correlated with the A parameter (10-item PSS: *r* = −0.14, *p* = 0.046; 14-item PSS: *r* = −0.15, *p* = 0.036). These results suggest that chronic stress is associated with more deontological decision-making, indicating that individuals with higher levels of chronic stress tend to make deontological moral judgments. These findings are generally consistent with the results of [47] ([47]).

Subsequently, the study conducted a reanalysis using the original continuous scores of stress and moral decision-making. As shown in Table 2, there was no statistically significant association between chronic stress and moral decision-making parameters. Additionally, the researchers reanalyzed the correlation between chronic stress and moral decision-making parameters using the 14-item Perceived Stress Scale scores (M = 39.14, SD = 6.43), with gender as a covariate. The results revealed no detectable correlation between chronic stress and overall action/inaction bias (*r* = −0.10, *p* = 0.187, Cohen’s *d* = −0.20, 1 − β = 0.40), consequence sensitivity (*r* = −0.06, *p* = 0.382, Cohen’s *d* = −0.12, 1 − β = 0.21), or norm sensitivity (*r* = 0.06, *p* = 0.443, Cohen’s *d* = 0.12, 1 − β = 0.21).

### 2.3. Discussion

The results obtained using the CAN algorithm indicate that there is no evidence of a correlation between chronic stress and moral decision-making, which is inconsistent with the original analysis. In the original results of [47] ([47]), the group with high chronic stress exhibited a stronger preference for inaction compared to the group with low chronic stress. However, this effect was not observed in the current reanalysis study. It is possible that this discrepant result may be attributed to the following two reasons.

Firstly, the I parameter generated by the CNI model is unreliable. [17] ([17]) hypothesized that a general tendency towards inaction or action would exist only when decision-makers do not consider consequences and norms. However, the lack of empirical support for this sequential processing assumption renders the I parameter unreliable ([12]; [29]). Therefore, the significant result obtained by previous studies may stem from methodological flaws with the I parameter of the CNI model.

Secondly, dichotomizing continuous data heightens the risk of Type I errors. The analysis shows a detectable negative correlation between chronic stress and action/inaction bias when using dichotomized data but not when using the original continuous data. As previous studies have found ([1]; [2]), dichotomizing continuous variables can amplify statistical errors, increasing the risk of false-positive results. In this study, similar results to prior studies are observed when dichotomized moral decision-making data are used. However, no significant associations are found between chronic stress and moral decision-making parameters when using the original continuous scoring data for calculations.

Furthermore, the correlation analysis using original scores revealed a small, non-significant correlation between chronic stress and CAN parameters. However, in the study conducted by [47] ([47]), there was a detectable difference in the scores of the general inaction preference between the high-stress group (M_High_ = 0.47, SD_High_ = −0.14) and the low-stress group (M_Low_ = 0.41, SD_Low_ = −0.14). The calculated Cohen’s *d* = 0.44, and 1 − β = 0.87, which indicates that the statistical power of the 197 samples in [47]’s ([47]) study was sufficient. These results suggest that the absence of a detectable correlation might not be due to sample size. Thus, for future replication studies, a similar sample size is advisable.

## 3. Study 2: The Replication and Extension of the Effect of Chronic Stress on Moral Decision-Making

### 3.1. Participants

The data for Study 2 were obtained through the Credamo Network Research Company, a professional research platform with a sample database encompassing all provincial-level administrative regions in China. After exclusions for recent acute stress, incomplete surveys, and failed attention checks, 208 valid responses were obtained (research materials and data are available at https://osf.io/693vn/?view_only=df880b9d19574c42b29bb29594cc7e00, accessed on 3 June 2024). The participants ranged from 17 to 89 years old (M = 32.07, SD = 12.35), with 90 males (43.3%) and 118 females (56.7%). All participants voluntarily participated in the study and received compensation after completing the questionnaire.

### 3.2. Materials and Methods

Study 2 was a within-subject dual-factor (moral decision-making and chronic stress) study, and the measurement materials for moral decision-making and chronic stress were consistent with previous studies ([47]; [17]). First, all participants underwent the measurement of the Perceived Stress Scale (PSS-10), which was revised by [44] ([44]). Next, all participants were required to make choices in 24 moral scenario stories. The moral decision-making materials were excerpted from the moral scenario stories compiled by [17] ([17]). The order of presentation of the scenarios was pre-randomized, ensuring that adjacent materials came from different stories. The behaviors in the 24 moral scenarios were divided into four categories: (1) norm-prohibited with benefits outweighing costs; (2) norm-prohibited with costs outweighing benefits; (3) norm-allowed with benefits outweighing costs; and (4) norm-allowed with costs outweighing benefits. Under each category, there were six different stories, and participants were required to choose whether the behavior described in the material was acceptable to them.

To avoid interference from other factors, this study controlled for social desirability using the abbreviated Marlowe–Crowne Scale ([37]) with 13 items. Participants responded with “yes” or “no,” and higher scores reflected greater reliance on social approval (Cronbach’s α = 0.76). As this study aimed to explore the impact of chronic stress on moral decision-making, the researchers designed specific questions (“Have you experienced an acute stressful event in the past week?”) at the beginning of the questionnaire to screen out participants who had recently faced acute stress events.

First, the researchers conducted the initial processing of participants’ responses (acceptable/unacceptable) under each scenario. Subsequently, the researchers analyzed the raw data through a series of analytical frameworks, including the traditional analysis, the process dissociation procedure ([10]), the CNI model ([25]), and the CAN algorithm ([29]), to derive participants’ moral decision-making tendencies under different analytical paradigms.

### 3.3. Results

#### 3.3.1. Traditional Analyses

The traditional analysis focuses on a single moral dilemma in which norms prohibit an action yet its benefits exceed costs, such as the trolley problem. In such dilemmas, accepting the proposal to act reflects utilitarianism, while rejection adheres to deontology. The participants’ utilitarian preferences are calculated based on their probability of accepting the action in six “norm-prohibited–benefits outweigh costs” scenarios.

The results reveal that, in the six “norm-prohibited–benefits outweigh costs” scenarios, the average probability of participants accepting the action was 0.62 (SD = 0.23). When setting 0.5 as the test value and performing a one-sample *t*-test, the analysis found that the probability of participants accepting the action was higher than 0.5 (*t* = 7.56, *p* < 0.001, *df* = 207, Cohen’s *d* = 0.52, 1 − β = 1.00). This indicates that, compared to rejecting the action, participants were more inclined to accept it, suggesting evidence for utilitarian decision-making preference. However, further analysis revealed that, after controlling for gender, age, and total social desirability scores as covariates, there was no reliable correlation between chronic stress and participants’ preferences in moral dilemma responses (*r* = 0.07, *p* = 0.326, *df* = 203).

#### 3.3.2. Process Dissociation Analyses

The process dissociation procedure advances from traditional analyses, proposing that utilitarian and deontological tendencies coexist. It categorizes norm-prohibited moral dilemmas into (1) inconsistent dilemmas, where benefits outweigh costs, and (2) consistent dilemmas, where costs outweigh benefits. Based on participants’ choices, four response types emerge: accepting/rejecting in inconsistent dilemmas and accepting/rejecting in consistent dilemmas. These responses enable the quantification of participants’ preferences for utilitarian and deontological decision-making ([10]).

The results show that the average degree of utilitarian-driven decision-making (U parameter) was 0.19 (SD = 0.28), and the average degree of deontological-driven decision-making (D parameter) was 0.47 (SD = 0.27), as shown in Table 3. Further analysis revealed that, after controlling for gender, age, and total social desirability scores as covariates, there was no detectable correlation between chronic stress and the U parameter (*r* = −0.02, *p* = 0.757, Cohen’s *d* = −0.04, 1 − β = 0.10). Similarly, there was no significant correlation between chronic stress and the D parameter (*r* = −0.09, *p* = 0.211, Cohen’s *d* = −0.18, 1 − β = 0.35).

#### 3.3.3. CNI Model Analyses

W employed the multiple-tree processing model ([25]) to obtain individual-level parameters: C (consequence sensitivity), N (norm sensitivity), and I (generalized inaction/action preferences). After excluding 40 participants with significantly poor model fit (*p* > 0.05) and 4 participants with error reports, 164 valid responses remained. Pearson’s correlation analysis showed that there was no significant associations among chronic stress, the C parameter (*r* = −0.09, *p* = 0.255, Cohen’s *d* = −0.18, 1 − β = 0.31), the N parameter (*r* = −0.12, *p* = 0.145, Cohen’s *d* = −0.23, 1 − β = 0.43), or the I parameter (*r* = −0.11, *p* = 0.179, Cohen’s *d* = −0.22, 1 − β = 0.39). The detailed results are presented in Table 4.

#### 3.3.4. CAN Algorithm Analyses

Through algebraic operations using the CAN algorithm, the values of the C parameter (consequence sensitivity), N parameter (norm sensitivity), and A parameter (overall Action/inaction bias) were obtained. Pearson’s correlation analysis revealed that there was no detectable correlation between chronic stress and the C parameter (*r* = −0.08, *p* = 0.229, Cohen’s *d* = −0.17, 1 − β = 0.33), the A parameter (*r* = 0.11, *p* = 0.124, Cohen’s *d* = 0.22, 1 − β = 0.46), or the N parameter (*r* = −0.05, *p* = 0.525, Cohen’s *d* = −0.09, 1 − β = 0.16). The detailed results are presented in Table 5.

Overall, the results indicate that there was no reliable evidence to show that chronic stress have a strong impact on individuals’ moral decision-making.

### 3.4. Discussion

The results of the replication study and the reanalysis of the data are consistent, indicating that chronic stress does not appear to enhance a preference for deontological decision-making. This finding contradicts the conclusions of previous studies ([47]).

The differences between this study and previous studies can be explained by the unreliability of the I parameter in the CNI model, as well as by the GAS theory and the stress process theory. While the dual-process model of moral judgment suggests that both emotional and cognitive processes play a role in moral judgments, they compete with each other ([19], [20]; [22], [21]; [24]; [32]). When the emotional process overcomes the cognitive process, individuals are driven by the emotional process and make deontological judgments. Conversely, when the cognitive process prevails over the emotional process, individuals are dominated by the cognitive process and make utilitarian judgments. However, when faced with chronic stress, according to the General Adaptation Syndrome theory and the stress process theory ([40]; [34]), individuals are able to utilize their resources to cope with and adapt to stress. In such cases, the cognitive and emotional processes in moral decision-making may be relatively balanced, and the individual’s moral decision-making is less likely to be dominated by either process. Therefore, chronic stress appears to exert minimal influence on moral decision-making.

Regarding participant selection, this replication study recruited participants randomly from the Credamo platform. This approach aimed to broaden the demographic diversity of the participant pool, thereby enhancing the generalizability of the finding and potentially reducing influences associated with specialized psychological knowledge or participant expectations.

Given the ethical nature of moral situations, social desirability may bias responses. Additionally, due to the complexity of modern life, individuals may concurrently experience acute and chronic stress. Acute stress, as a primary stressor, can trigger secondary stress, leading to simultaneous acute and chronic stress. And acute stress tends to increase deontological decision-making ([28]). Therefore, it is necessary to exclude individuals who are experiencing acute stress from the study. This study included specific questions and social desirability scales to eliminate the influence of social desirability and acute stress on the results, thus improving the validity of the research conclusions.

## 4. General Discussion

In this study, the researchers not only reanalyzed the existing research data but also conducted a replication study by collecting new data and applying multiple analysis approaches. The results consistently indicate that chronic stress does not lead to more deontological judgments.

In summary, this study found no detectable correlation between chronic stress and individuals’ consequence sensitivity, norm sensitivity, or overall action/inaction preferences. This suggests that chronic stress exerts minimal influence on moral decision-making. This conclusion clarifies the effect of chronic stress on moral decision-making and provides empirical evidence for re-evaluating its influence, urging caution when using the CNI model for moral judgment research ([12]; [13]; [31]).

This conclusion can be partly supported by the GAS theory and the Stress Process Model ([40]; [34]). Firstly, when faced with chronic stress, individuals tend to demonstrate more adaptability ([36]). The organism can fully utilize personal resources to cope with chronic stress, thereby mitigating its negative impact and enabling them to maintain a relatively calm mind and sufficient cognitive resources, rather than falling into an irrational or emotional state of high stress. Therefore, in a state of chronic stress, individual adaptability weakens the impact of stress on moral judgment.

From a practical standpoint, these findings suggest that individuals’ moral decision-making may be a relatively stable trait or cognitive pattern that is not readily influenced by chronic stress. Moreover, this conclusion indicates that prior researchers may have overemphasized the impact of stress on moral decision-making while neglecting other underlying factors, such as personality traits (such as the trait anxiety and the trait anger) and physiological factors ([11]; [39]). Future research could further explore the effects of acute and chronic stress on moral decision-making, including other variables induced by stress, such as changes in hormone levels. While this study has further demonstrated the effect of chronic stress on moral decision-making through a series of methods and theories, there are still some limitations that require improvement in subsequent research. Firstly, the PSS-10 measurement of chronic stress may be inaccurate due to self-reporting biases. To enhance accuracy, future research could adopt hair cortisol as a chronic stress indicator, preferred for its sampling, measurement, and analytical advantages ([38]). Moreover, the PSS-10 is designed to assess the stress experienced by participants over the past four weeks ([44]), meaning that the scale only represents the stress level within the preceding month. However, the GAS theory emphasizes that adaptation to stress is a long-term, gradual process. It remains unclear whether the chronic stress experienced by individuals will diminish or intensify over time. Therefore, the level of chronic stress obtained through the PSS-10 can only serve as a reference index of chronic stress. Whether more prolonged chronic stress (e.g., lasting for several months or years) exerts a more profound influence on individuals’ moral dilemma decision-making remains to be further explored.

Secondly, to ensure consistency with the original study, Study 2 used 24 moral scenario stories. However, [25] ([25]) suggested using 48 scenarios for accurate individual-level CNI model application. This richer set enhances the stability of C, N, and I parameters. Hence, future research employing the CNI model to analyze individual moral judgment differences should adopt extended 48 moral dilemma stories recommended by [25] ([25]).

In addition, the lack of a significant utilitarian or deontological moral decision-making tendency among individuals under chronic stress may also be related to the heterogeneity of individual physiological responses. Some studies have shown that lower baseline cortisol levels are associated with decision-making dysfunction, but not all individuals under chronic stress exhibit this effect, indicating that there may be individual differences in physiological regulation ([7]; [26]). However, this factor was not controlled for in the present study.

Lastly, individuals in daily life often face multiple concurrent stress sources. For instance, unemployment can strain financial and marital situations, with unemployment serving as a primary stressor that triggers subsequent stressful stimuli. This interconnectedness between stressors is referred to as “stress proliferation” by researchers ([33]). However, the present study only examined chronic stress as a singular stressor, without clarifying whether it is secondary to other factors or a primary stressor leading to further stressful stimuli. Therefore, future research can further explore the transmission effects of stress, clarifying which stressor specifically affects moral decision-making and what subsequent impacts it brings.

## 5. Conclusions

In summary, this study reanalyzed existing data on chronic stress and moral decision-making and conducted a replication study, finding that there is no evidence for a significant correlation between chronic stress and moral decision-making. Chronic stress does not significantly increase deontological decision-making tendencies.

## Figures and Tables

**Table 1 behavsci-15-01068-t001:** Partial correlations between chronic stress and dichotomized moral decision data in Study 1 (n = 197).

	*M* ± *SD*	1	2	3	4	
1. PSS-10	26.51 ± 5.33	1				
2. PSS-14	39.14 ± 6.43	0.96 ***				
3. Consequence sensitivity	0.16 ± 0.20	−0.02	−0.02	1		
4. Norm sensitivity	0.10 ± 0.29	0.04	0.02	0.08	1	
5. Overall action/inaction bias	0.55 ± 0.14	−0.14 *	−0.15 *	−0.27 ***	0.06	1

Note. The moral decision indices are computed with the dichotomized data. Gender is controlled. * *p* < 0.05; *** *p* < 0.001.

**Table 2 behavsci-15-01068-t002:** Partial correlations between chronic stress and moral decision indices in Study 1 (n = 197).

	*M* ± *SD*	1	2	3	4
1. Chronic stress	26.51 ± 5.33	1			
2. Consequence sensitivity	0.46 ± 0.58	−0.06	1		
3. Norm sensitivity	0.36 ± 0.86	0.06	0.22 **	1	
4. Overall action/inaction bias	3.67 ± 0.41	−0.10	−0.15 *	−0.03	1

Note. Chronic stress is measured by the 10-item version of the PSS. The moral decision indices are computed with the original moral acceptability rating scores. Gender is controlled. * *p* < 0.05; ** *p* < 0.01

**Table 3 behavsci-15-01068-t003:** Partial correlations between the U parameter, the D parameter, the probability of utilitarian action, and chronic stress (n = 208).

	*M* ± *SD*	1	2	3	4
1. chronic stress	12.15 ± 6.84	1			
2. U parameter	0.19 ± 0.28	−0.02	1		
3. D parameter	0.47 ± 0.27	−0.09	−0.03	1	
4. probability of utilitarian action	0.62 ± 0.23	0.07	0.47 ***	−0.80 ***	1

Note. Chronic stress is measured by the 10-item version of the PSS. Gender, age, and social desirability are controlled. *** *p* < 0.001.

**Table 4 behavsci-15-01068-t004:** Partial correlations between the C parameter, the N parameter, I parameter, and chronic stress (n = 164).

	*M* ± *SD*	1	2	3	4
1. chronic stress	12.33 ± 6.86	1			
2. C parameter	0.19 ± 0.20	−0.09	1		
3. N parameter	0.25 ± 0.27	−0.12	0.02	1	
4. I parameter	0.41 ± 0.24	−0.11	−0.04	−0.10	1

Note. Chronic stress is measured by the 10-item version of the PSS. Gender, age, and social desirability are controlled.

**Table 5 behavsci-15-01068-t005:** Partial correlations between the C parameter, A parameter, N parameter, and chronic stress (n = 208).

	*M* ± *SD*	1	2	3	4
1. chronic stress	12.15 ± 6.84	1			
2. C parameter	0.20 ± 0.21	−0.08	1		
3. A parameter	0.55 ± 0.10	0.11	−0.22 ***	1	
4. N parameter	0.06 ± 0.34	−0.05	−0.08	0.07	1

Note. Chronic stress is measured by the 10-item version of the PSS. Gender, age, and social desirability are controlled. *** *p* < 0.001.

## Data Availability

The data that support the findings of this study are openly available via https://osf.io/693vn/?view_only=df880b9d19574c42b29bb29594cc7e00 (accessed on 3 June 2024).

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
