# Peer review of "No Detectable Effects of Chronic Stress on Moral Decision-Making Are Found: A Data Reanalysis and a Pre-Registered Replication and Extension"

_behavsci, 2025, doi:10.3390/bs15081068_

Round 1

Reviewer 1 Report

Comments and Suggestions for Authors

This study investigates whether chronic (long-term) stress influences moral decision-making. Prior research by Zhang et al. (2018) suggested that individuals experiencing chronic stress are more likely to make deontological (rule-based) moral judgments—for example, refusing to harm one person even if it would save others. However, the current authors question the reliability of this effect, arguing that people tend to adapt to prolonged stress over time, which may minimize its impact on moral reasoning.

To test this, the authors reanalyzed the original data and conducted a new replication study using improved methods and a more diverse sample. Both studies found no evidence that chronic stress affects moral decision-making. The authors conclude that earlier findings were likely due to methodological flaws, and that chronic stress does not significantly change how people evaluate moral dilemmas. The study was well designed, with a clear motivation and appropriate statistical analyses. However, I have the following concerns:

  • The study relies entirely on self-report measures for both chronic stress (PSS-10) and moral decision-making (hypothetical scenarios). While the authors mention limitations of self-reported stress, they do not address the broader issue that both key variables are subjective and may be influenced by social desirability or perceived expectations. Without behavioral or physiological data, it’s unclear whether the findings reflect real-life moral decision-making under stress. I suggest the authors discuss this limitation further or, if possible, include additional experimental evidence.

  • The paper suffers from a critical conceptual flaw in how it defines and measures "chronic stress." The authors use the PSS-10, which measures perceived stress over the past month, but then theoretically justify their null findings using General Adaptation Syndrome (GAS) theory. This creates a fundamental mismatch: GAS theory specifically refers to prolonged exposure to stressors over extended periods (months to years), not the 4-week timeframe captured by PSS-10. The authors can try to find better theoretical support for their use of PSS-10, or they should address this limitation in Discussion.

Author Response

Comments from Reviewer #1:

This study investigates whether chronic (long-term) stress influences moral decision-making. Prior research by Zhang et al. (2018) suggested that individuals experiencing chronic stress are more likely to make deontological (rule-based) moral judgments—for example, refusing to harm one person even if it would save others. However, the current authors question the reliability of this effect, arguing that people tend to adapt to prolonged stress over time, which may minimize its impact on moral reasoning.

To test this, the authors reanalyzed the original data and conducted a new replication study using improved methods and a more diverse sample. Both studies found no evidence that chronic stress affects moral decision-making. The authors conclude that earlier findings were likely due to methodological flaws, and that chronic stress does not significantly change how people evaluate moral dilemmas. The study was well designed, with a clear motivation and appropriate statistical analyses. However, I have the following concerns:

  1. The study relies entirely on self-report measures for both chronic stress (PSS-10) and moral decision-making (hypothetical scenarios). While the authors mention limitations of self-reported stress, they do not address the broader issue that both key variables are subjective and may be influenced by social desirability or perceived expectations. Without behavioral or physiological data, it’s unclear whether the findings reflect real-life moral decision-making under stress. I suggest the authors discuss this limitation further or, if possible, include additional experimental evidence.

RE: Thank you for your suggestions. The present study utilized the PSS-10 to measure chronic stress primarily to ensure the rigor of replication by maintaining consistency with the materials used in the study by Zhang et al. (2018). However, it must be acknowledged that relying solely on self-reports for the measurement of chronic stress may indeed introduce bias. Nevertheless, considering the widespread use of the PSS-10 in assessments, it can serve as a reference index of chronic stress. Therefore, we have, on the one hand, incorporated additional theoretical and empirical evidence in the main text to elucidate the impact of chronic stress on individuals (e.g., p2, “Folkman (2013) further emphasized that the stress process constitutes a dynamic system in which responses to stressors are moderated by situational difficulty, personal resources, and personality traits. Building on extant theory, Gordon et al. (2021) demonstrated empirically that stress responses are jointly determined by external demands and personal resources. External demands denote challenges and tasks that function as potential stressors, whereas resources encompass internal or external supports, such as emotion-regulation capacity, social support, and financial assets, which are available for coping with those demands. A higher demand-to-resource ratio (i.e., resource insufficiency) significantly amplifies perceived stress, as indexed by elevated blood pressure (Gordon et al., 2021)”). On the other hand, we have further elaborated on the limitations of the PSS-10 as an indicator of chronic stress in the discussion (e.g., p10, “In addition, the lack of a significant utilitarian or deontological moral decision-making tendency among individuals under chronic stress may also be related to the heterogeneity of individual physiological responses. Some studies have shown that lower baseline cortisol levels are associated with decision-making dysfunction, but not all individuals under chronic stress exhibit this effect, indicating that there may be individual differences in physiological regulation (Ceccato et al., 2016; Landolt et al., 2017). However, this factor was not controlled for in the present study”).

  1. The paper suffers from a critical conceptual flaw in how it defines and measures “chronic stress”. The authors use the PSS-10, which measures perceived stress over the past month, but then theoretically justify their null findings using General Adaptation Syndrome (GAS) theory. This creates a fundamental mismatch: GAS theory specifically refers to prolonged exposure to stressors over extended periods (months to years), not the 4-week timeframe captured by PSS-10. The authors can try to find better theoretical support for their use of PSS-10, or they should address this limitation in Discussion.

RE: Thank you for bringing this limitation to our attention. Selye’s (1950) General Adaptation Syndrome theory indeed focuses on long-term, intractable stressors (such as chronic illness, persistent poverty, and ongoing work stress). According to the GAS theory, individuals under chronic stress will enter the alarm stage within a relatively short period of time, subsequently experience the resistance stage over days, weeks, or months, and finally, if their bodily resources are exhausted, they will enter the exhaustion stage, which may last for months to years. However, the PSS-10 used in this study only allows participants to evaluate their stress levels over the past four weeks. Therefore, we cannot determine which stage of chronic stress the participants are experiencing, nor can we determine whether the stages they have experienced reflect the entire process of the GAS theory. However, the core tenets of both the GAS theory and the stress process framework emphasize the adaptive process of individuals to stress. Thus, this study can be partially supported by these two theories, and we further pointed out the limitations of the measurement in this study in the discussion (e.g., p9, “Moreover, the PSS-10 is designed to assess the stress experienced by participants over the past four weeks (Wang et al., 2011), meaning that the scale only represents the stress level within the preceding month. However, the GAS theory emphasizes that adaptation to stress is a long-term, gradual process. It remains unclear whether the chronic stress experienced by individuals will diminish or intensify over time. Therefore, the level of chronic stress obtained through the PSS-10 can only serve as a reference index of chronic stress. Whether more prolonged chronic stress (e.g., lasting for several months or years) exerts a more profound influence on individuals' moral dilemma decision-making remains to be further explored”).

Reviewer 2 Report

Comments and Suggestions for Authors

Thank you very much for allowing me to review this exciting manuscript. In my opinion, a better understanding of the links between chronic stress and moral decisions has important implications for science and practice.  These implications (especially for practice) could be elaborated on a little more by the authors. 

I have a few comments:

- In Study 1, the relevance of the reanalysis of the data could be made even clearer.

- Regarding the overall theoretical framework: it might be useful to also address the discourse on the topic of “moral intuitionism vs. rationalism” and discuss the results of the studies against this backdrop and taking into account additional parameters such as personality.

- I also noticed a few grammatical errors. Perhaps a language edit would be helpful?

- I would also advise against (and here I mean in terms of wording) focusing so much on the contradiction between your own analyses and the results of Zhang et al. (2018) (cf. first sentence of the discussion). Instead, I would prefer to argue on the basis of content. Otherwise, it appears that the aim of the studies is to refute Zhang et al. (2018) rather than to gain new insights.

- The authors state "Chronic stress does not significantly increase deontological decision-making tendencies" in the end of the conclusion, but what does this mean in practice? 

Author Response

Comments from Reviewer #2:

Thank you very much for allowing me to review this exciting manuscript. In my opinion, a better understanding of the links between chronic stress and moral decisions has important implications for science and practice.  These implications (especially for practice) could be elaborated on a little more by the authors. 

I have a few comments:

  1. In Study 1, the relevance of the reanalysis of the data could be made even clearer.

RE: Thank you for your suggestions. We further refined the data analysis section of Study 1, supplementing the complete results of the correlation analysis between chronic stress and CAN parameters derived from dichotomized data, and presented them in tabular form.

  1. Regarding the overall theoretical framework: it might be useful to also address the discourse on the topic of “moral intuitionism vs. rationalism” and discuss the results of the studies against this backdrop and taking into account additional parameters such as personality.

RE: Thank you for your insightful suggestions. Moral intuitionism and moral rationalism are closely related to moral dilemmas. However, this aspect overlaps theoretically with the dual-process model of moral judgment proposed by Greene et al. (2001, 2008), which posits that individuals may possess both moral intuitive reasoning (dominated by emotional processes) and moral rational reasoning (dominated by cognitive processes), with both competing and functioning in moral judgments. The dual-process model of moral judgment by Greene et al. (2001, 2008) has already been discussed in the main text (e.g., p8, “While the dual-process model of moral judgment suggests that both emotional and cognitive processes play a role in moral judgments, and they compete with each other (Greene, 2007; Greene, 2009; Greene et al., 2001; Greene et al., 2008; Koenigs et al., 2007; Paxton & Greene, 2010). When the emotional process overcomes cognitive process, individuals are driven by emotional process and make deontological judgments. Conversely, when the cognitive process prevails over emotional process, individuals are dominated by cognitive process and make utilitarian judgments”), and therefore, we did not add further discussion on moral rationalism and moral intuitionism. Nevertheless, personality factors, such as trait anxiety and trait anger, do indeed influence individuals' responses to chronic stress (Duque et al., 2022; Schneiderman et al., 2005). Thank you again for your suggestion. This point has been supplemented in the discussion section (e.g., p9, “Moreover, this conclusion indicates that prior researchers may have overemphasized the impact of stress on moral decision-making while neglecting other underlying factors, such as personality traits (such as trait anxiety and trait anger) and physiological factors”).

  1. I also noticed a few grammatical errors. Perhaps a language edit would be helpful?

RE: Thank you for your reminder. We have once again carefully checked the grammar of the entire text to ensure that the language and grammar are correct.

  1. I would also advise against (and here I mean in terms of wording) focusing so much on the contradiction between your own analyses and the results of Zhang et al. (2018) (cf. first sentence of the discussion). Instead, I would prefer to argue on the basis of content. Otherwise, it appears that the aim of the studies is to refute Zhang et al. (2018) rather than to gain new insights.

RE: Thank you for your kind advice. We have adjusted the wording of the relevant content in the main text to ensure objective and impartial language. Our expression now focuses more centrally on the substance of this study, rather than overemphasizing the rebuttal of Zhang et al.'s (2018) research. Thank you again for your kind suggestions.

  1. The authors state "Chronic stress does not significantly increase deontological decision-making tendencies" in the end of the conclusion, but what does this mean in practice? 

RE: Thank you for your question. We acknowledge that the original discussion in the manuscript was insufficient in the practical implications. In the revised discussion section, we have supplemented this aspect. On the one hand, the findings suggest that individual moral decision-making may represent a relatively stable trait or behavioral pattern, not easily altered by external factors. On the other hand, previous research appears to have overemphasized the impact of stress on moral decision-making while overlooking other influential factors (e.g., personality traits, biological factors). The current results call attention to the need to examine not only the direct effects of stress itself on moral decision-making but also other underlying factors associated with the stress experience.